# INFERENCE-TIME PERSONALIZED FEDERATED LEARNING

## ABSTRACT

In Federated learning (FL), multiple clients collaborate to learn a model through a central server but keep the data decentralized. Personalized federated learning (PFL) further extends FL to handle data heterogeneity between clients by learning personalized models. In both FL and PFL, all clients participate in the training process and their labeled data is used for training. However, in reality, novel clients may wish to join a prediction service after it has been deployed, obtaining predictions for their own unlabeled data.

Here, we defined a new learning setup, Inference-Time PFL (IT-PFL), where a model trained on a set of clients, needs to be later evaluated on novel unlabeled clients at inference time. We propose a novel approach to this problem IT-PFL-HN, based on a hypernetwork module and an encoder module. Specifically, we train an encoder network that learns a representation for a client given its unlabeled data. That client representation is fed to a hypernetwork that generates a personalized model for that client. Evaluated on four benchmark datasets, we find that IT-PFL-HN generalizes better than current FL and PFL methods, especially when the novel client has a large domain shift. We also analyzed the generalization error for the novel client, showing how it can be bounded using results from multi-task learning and domain adaptation. Finally, since novel clients do not contribute their data to training, they can potentially have better control over their data privacy; indeed, we showed analytically and experimentally how novel clients can apply differential privacy to their data.

## 1 INTRODUCTION

Personalized Federated learning (PFL) Zhao et al. (2018) extends federated learning (FL) McMahan et al. (2017a) to the case where the data distribution varies across clients. This has numerous applications like when a smartphone application wishes to improve their text prediction without uploading user-sensitive data, or in the case when a consortium of hospitals wishes to train a joint model while preserving the privacy of their patients. Current methods assume that all clients participate in the training process and that their data is labeled (Shamsian et al., 2021; Bui et al., 2019; Hsu et al., 2020; Zhu et al., 2020; Yang et al., 2020; Hanzely & Richtárik, 2020), so once a model is trained, there is no way to add novel clients.

In many cases, however, a federated model has been trained and deployed, but then novel clients wish to join. Often, such novel clients have no labeled data, and the distribution of their samples may shift significantly. This is the case for example, when a speech-recognition federated model has been deployed and needs to be applied to new users, or when a virus diagnostic has been developed for some regions or countries, and then needs to be applied to new populations while the virus spreads. This learning setup poses a hard challenge to existing FL approaches. FL techniques may not generalize well to novel clients with different data distribution. PFL techniques learn personalized models but are not designed to apply to a client that was not available during training.

Here, we define a novel problem, performing federated learning in novel clients with unlabeled data that are only available at inference time. We call this setup **IT-PFL** for *Inference-Time Personalized Federated Learning*. We propose a novel approach to this problem, called IT-PFL-HN. During the training phase, our architecture learns a space of personalized models, one for each client, together with an encoder that maps each client to a point in this space. All personalized models are learned

jointly through a hypernetwork, allowing to use personalized data effectively (Figure 1). At inference time, a novel client can locally compute its own descriptor using the encoder. Then is sends the descriptor to the server as input to the hypernetwork to obtain its personalized model.

The key idea in IT-PFL-HN is to first train a client encoder, that maps a full client dataset into a dense descriptor. That descriptor is fed into a hypernetwork, which learns to map client descriptors onto a space of networks. At inference time, any novel client simply computes it descriptor locally using the encoder. It then sends that descriptor to the server, which sends back an "on-demand" personalized model for that client.

FL was motivated by privacy, but was shown vulnerable to differential attacks (Xie et al., 2018; Augenstein et al., 2019; Melis et al., 2019; Zhu & Philip, 2019; Hao et al., 2019; Truex et al., 2019; Mothukuri et al., 2021; Hitaj et al., 2017). In our setup of IT-PFL, novel clients do not contribute their data to training, so they have better control over their data privacy. We show how differential privacy (DP) can be applied effectively to a novel client and experimentally measured how applying DP to the client affects the accuracy of the personalized model.

This paper makes the following contributions: (1) A new learning setup, IT-PFL, learning a personalized model to novel unlabeled clients at inference time. (2) A new approach, learning a space of models and an encoder that maps a client to that space, and architecture IT-PFL-HN, based on hypernetworks. (3) A bound on the generalization error, based on MTL and DA. (4) Analysis of differential privacy for the novel client. (5) Evaluation on 4 benchmark datasets, *CIFAR10*, *CIFAR100*, *iNaturalist*, *Landmarks*, showing that IT-PFL-HN performs better or equally well as baselines.

## 2 RELATED WORK

**Federated learning (FL)** In FL, clients collaboratively solve a learning task. The key motivation for individual clients to participate in FL is to leverage the shared pool of knowledge from other clients in the federation. Individual clients often face data constraints such as data scarcity, low data quality, and unseen classes that limit their capacity to train well performing local models. FedAvg (McMahan et al., 2017a), is an early but effective FL approach that updates models locally and averages them into a global model. Several optimization methods have been proposed for improving convergence in FL (Sahu et al., 2018; Lin et al., 2018; Stich, 2018; Wang & Joshi, 2018). Other approaches focus on preserving client privacy (Agarwal et al., 2018; Duchi et al., 2014; McMahan et al., 2017b; Zhu et al., 2020), improving robustness to statistical diversity (Haddadpour & Mahdavi, 2019; Hanzely & Richtárik, 2020; Hsu et al., 2019b; Karimireddy et al., 2020; Zhao et al., 2018; Zhou & Cong, 2017), and reducing communication cost (Reisizadeh et al., 2020; Dai et al., 2019). These methods aim to learn a global model across clients, which limits their ability to deal with heterogeneous clients.

**Personalized federated learning (PFL)** aims to handle data heterogeneity across clients. Following Tan et al. (2021), here we divide PFL methods into data-based and model-based approaches.

Data-based approaches aim to smooth the statistical heterogeneity of data among different clients. This can be achieved by normalizing the data (Duan et al., 2021), or by designing client selection mechanisms that enable sampling from a more homogeneous data distribution. Wang et al. (2020b) selects a subset of participating clients for each training round, with the objective of maximizing accuracy while minimizing the number of communication rounds. Yang et al. (2020) selects the subset of clients with minimal class imbalance.

Model-based approaches aim to enable FL models to adapt to the diverse data distributions among clients. Jiang et al. (2019) and Fallah et al. (2020) use meta-learning for finding a global model that is efficiently optimized with few steps of gradient descent using the client local data. Achituve et al. (2021) propose learning a single kernel function shared by all clients, but use a personalized Gaussian processes classifier for each client. Bui et al. (2019) and Liang et al. (2020b) achieve personalization by using a global model, and a local model on top of it. The local model does not share parameters with the server. Huang et al. (2021) regularized stronger collaboration amongst clients with similar data distributions.

Figure 1: **The architecture of IT-PFL.** The encoder uses the data of the novel client to produce an embedding. Then, the hypernetwork $f_\theta$ predicts a local model $h$ for the novel client.

Adapting a model to a new distribution at inference time was also considered as a variant of domain adaptation (DA) (Wang et al., 2020a; Kim et al., 2021; Liang et al., 2020a). Our new client setup can be seen as an adaptation of this setup to the federated setup.

**Differential privacy (DP)** The goal is to share information about a dataset while withholding information about individuals in the dataset. Although one of the key motivations of PFL is privacy, it has been shown that some private information is exposed in the process (Mothukuri et al., 2021). Melis et al. (2019) shows that an adversarial participant can infer the presence of exact data points in others training data. Hitaj et al. (2017) shows that an adversarial participant can generate other clients private data. Several works utilizes DP to protect the client privacy (Xie et al., 2018; Augenstein et al., 2019; Zhu & Philip, 2019; Hao et al., 2019; Truex et al., 2019). However, adding noise to the embedding of the clients may harm the model performance. Unlike these papers, we focus on the privacy of the novel client, so we do not harm the training process, and each novel client can choose its privacy-accuracy balance in real-time.

## 3 THE LEARNING SETUP

We now formally define the learning setup of IT-PFL.

Following the notation in Baxter (2000) we define $X$ to be an input space and $Y$ an output space. $P$ is a probability distribution over the data $X \times Y$. Let $l$ be a loss function $l : Y \times Y \to R$. $\mathcal{H}$ is a set of hypotheses $h : X \to Y$. The error of a hypothesis $h$ over a distribution $P$ is $err_P(h) = \int_{X \times Y} l(h(x), y) dP(x, y)$.

In IT-PFL, we are given a federation of $N$ clients $c_1, \ldots, c_N$ for training, and other, novel, clients added at inference time. For simplicity of notation, we consider a single novel client $c_{new}$. Let $\{P_i\}_{i=1}^N$ be the data distributions of training clients, and $P_{new}$ the data distribution for $c_{new}$. Each training client has access to $m_i$ IID samples from its distribution $P_i$, $S_i = \{(x_j^i, y_j^i)\}_{j=1}^{m_i}$.

The goal of IT-PFL is to use data from training clients $\{S_i\}_{i=1}^N$ to learn a mechanism that can assign a hypothesis $h_{new} \in \mathcal{H}$ when given data from a novel client $S_{new}$. That hypothesis should minimize the error on the novel client $err_{P_{new}}(h_{new})$.

## 4 OUR APPROACH

In IT-PFL, we are evaluated by the quality of inference on a novel client, one that was never seen during training, and has its own data distribution. Fundamentally, this problem is not about producing a single trained model, but about producing a "meta mechanism" that can provide a model "on-demand", given (a descriptor of) a novel client.

A natural candidate for such a meta-mechanism, are hypernetworks (HNs). HNs are neural networks that output the weights of another network, conditioned on some input, and can therefore produce "on-demand" models. Since the weights of the generated model are a (differentiable) function of the HN parameters, training the HN is achieved simply by propagating gradients from the generated (client) model. HNs have already been shown to be effective for PFL by Shamsian et al. (2021).

To generate a model for a novel client, the HN should be fed with a proper descriptor. We propose to learn an *Encoder* that takes as input the data of the novel client and produces a dense descriptor. The encoder learns an embedding space over clients. This allows to interpolate to novel clients, since the hypernetwork learns to produce a proper model for a client, based on its embedding.

(a) Training with clients 1..N:

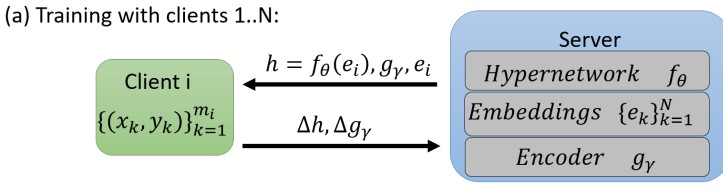

(b) Inference for a new client:

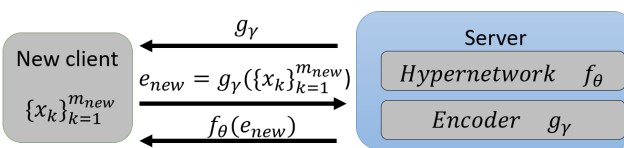

Figure 2: (a) The server trains a hypernetwork $f_\theta$ and an encoder $g_\gamma$ using the labeled clients. (b) A novel client uses unlabeled data to receive a personalized model from the server.

Figure 1 shows the model architecture, as used during inference. We now describe more formally the two model components: (1) hypernetwork and (2) encoder, then describe the flow during training and inference.

**Client encoding.** The HN expects to receive as input a descriptor of a client. The Encoder module takes unlabeled data and produces a dense descriptor. Consider a representation space $\mathcal{E}$ of clients, we aim to obtain a representation $e_i \in \mathcal{E}$ for client $i$. This is achieved by applying a deep-set architecture (Zaheer et al., 2017) to data batches, producing a descriptor for each batch. A single client descriptor is obtained by averaging all batch descriptors. One potential extension of our method is that one could apply this technique to individual batches, obtaining a personalized model per batch.

**Hypernetworks**. Assume first that there already exist a representation space $\mathcal{E}$ of clients, where a client $i$ has a representation $e_i \in \mathcal{E}$. This representation is used as an input to the HN, which then produces a personalized model for the client. We explain below how the representation is learned.

A hypernetwork $f_\theta$ parametrized by $\theta$ embodies a mapping from client-embedding space to hypotheses (model) space $f_\theta : \mathcal{E} \to H$. Any client with an embedding vector $e_i$ is mapped by the hypernetwork to a personalized model $h_i = f_\theta(e_i)$.

**Training** The hypernetwork learns to produce a personalized model from a client descriptor by optimizing the following loss $L_{HN}(\theta, e_1, ..., e_N) = \sum_{i=1}^{n} \sum_{j=1}^{m_i} l(f_\theta(e_i)(x_j^i), y_j^i)$. using training (labeled) clients $c_1, ..., c_N$.

When training the encoder, we found that in practice it worked better to learn embedding for the training clients, which takes a client identity and produces a dense descriptor. Presumably, this is because it reduces the batch-to-batch variability for batches from the same client, since all samples from a client $i$ receive a unique client descriptor. The embedding layer was trained in a standard way end-to-end with the hypernetwork. Once every client has an embedding descriptor $e_i$, we trained the client encoder, as described above, taking a data batch and producing a descriptor that is $L_2$ close to $e_i$. More details are given in appendix C.

**Flow** Figure 2 illustrates the steps of the federated training. During training, the hypernetwork $f_\theta$ optimizes $L_{HN}$ and the encoder $g_\gamma$ optimizes $L_{encoder} = \sum_{i=1}^{n} L_2(g_\gamma(\{x_j^i\}_{j=1}^{m_i}), e_i)$. Specifically, first, the server applies the hypernetwork to each client embedding $e_i$. Then, the server sends to each client $c_i$ its personalized model $h_i$ together with the encoder $g_\gamma$. The client trains the predicted network locally using its labeled data. Then, the client sends to the server its updates to the parameters of the network $h_i$, and to the encoder $g_\gamma$.

At inference time, the server only sends the encoder to the novel client. The client uses the encoder to calculate an embedding $\hat{e}_{new}$ and sends it to the server. The server uses the hypernetwork to predict the client personalized model $h_{new}$ from the embedding and sends the result to the client. The client then applies its personalized model locally without revealing its data.

## 5 GENERALIZATION BOUND

We now develop a bound for the generalization error of a novel client in the IT-PFL setup. We build on previous bounds developed for multi-task learning (MTL) and for domain adaptation (DA).

**Theorem 5.1.** *Let $\mathcal{H}$ be hypotheses space, $P_{new}$ be the data distribution of a novel client, and $Q$ be a distribution over client data distributions, namely $P_i$ is drawn from $Q$.*

*The generalization error of the novel client is bound by $err_{P_{new}}(H) \leq e\hat{r}r_z(H) + \epsilon + \frac{1}{2}\int_P \inf_{h \in H} \hat{d}_{H \Delta H}(P, P_{new})dQ(P)$. Here, $\epsilon$ is the approximation error of a client in the federation from Theorem 2 in Baxter (2000) and $\hat{d}_{H \Delta H}$ is the distance measure between probability distributions defined in Ben-David et al. (2010). The error $err_Q(H)$ and the empirical error $e\hat{r}r_z(H)$ are defined in detail in appendix A.*

*Proof.* See appendix A for the detailed proof. The main idea of the proof is to first bound the error of all (labeled) training clients, using results from multi-task learning. Then, treat the novel client as a target domain in a domain adaptation problem, and bound its shift from training clients.

The error of the novel client for a given hypothesis space $H$ is $\inf_{h \in H} err_{P_{new}}(h)$. Since $P_{new}$ is independent of $Q$, we can integrate over all $P \sim Q$ and obtain

$$err_{P_{new}}(H) := \inf_{h \in H} err_{P_{new}}(h) = \int_P \inf_{h \in H} err_{P_{new}}(h)dQ(P). \tag{1}$$

Using Theorem 2 from Ben-David et al. (2010) with $P_{new}$ treated as the target domain and $P$ as the source domain, gives that $\forall h, \forall P : err_{P_{new}}(h) \leq err_P(h) + \frac{1}{2}\hat{d}_{H \Delta H}(P, P_{new})$. Plugging into Eq. (1) gives

$$
\begin{aligned}
err_{P_{new}}(H) &\leq \int_P \inf_{h \in H} \left[ err_P(h) + \frac{1}{2}\hat{d}_{H \Delta H}(P, P_{new}) \right] dQ(P) \\
&= err_Q(H) + \frac{1}{2}\int_P \inf_{h \in H} \hat{d}_{H \Delta H}(P, P_{new})dQ(P).
\end{aligned}
\tag{2}
$$

Since $err_Q(H)$ is unknown, we use Theorem 2 from Baxter (2000) to bound the error of the novel client. That yields the bound in the theorem. □

## 6 EXPERIMENTS

We evaluate the empirical performance of IT-PFL-HN using four benchmark datasets.

### 6.1 EXPERIMENT SETUP AND EVALUATION PROTOCOL.

We evaluate IT-PFL-HN on the proposed IT-PFL setup where novel clients are presented to the server at inference time. **Client split:** To quantify performances on novel clients we first randomly partition clients to $N_{train}$ train clients and $N_{novel}$ novel clients. We used $N_{novel} = 0.1 \cdot N$. Unless stated otherwise, We report the average accuracy over novel clients: $\frac{1}{N_{novel}} \sum_i^{N_{novel}} \frac{1}{m_i} \sum_j^{m_i} Acc(\psi_i(x_j^i), y_j^i)$. Where $\psi_i$ is the model chosen by the server to evaluate novel client $i$ with $m_i$ samples. To conduct a fair comparison, training is limited to 500 steps for all evaluated methods. In each step, the server communicates with a $0.1$ fraction of training clients according to the protocol of each method. Specific model architectures are described for each experiment separately. **Sample split and HP tuning:** We split the samples of each training client into a training set and validation set. Validation samples were used for hyperparameter tuning for all methods and datasets. Specifically, we tuned training batch size, learning rate of each method, weight-decay values and the number of inner training epochs, and an early stopping point. Results are reported using the best hyperparameters for each experiment. See appendix C for more details.

### 6.2 BASELINES

We evaluate and compare the following FL and PFL methods: (1) **IT-PFL-HN**, our proposed IT-PFL method using HN and client encoder. Each novel client is evaluated using its own personalized

Table 1: **Accuracy on novel Clients, CIFAR Data:** Values are averages and standard error across clients.

| split | CIFAR-10 | | | | CIFAR-100 | | | |
|---|---|---|---|---|---|---|---|---|
| | pathological | $\alpha = 0.1$ | $\alpha = 0.5$ | $\alpha = 10$ | pathological | $\alpha = 0.1$ | $\alpha = 0.5$ | $\alpha = 10$ |
| FedAvg | $50.3 \pm 2.9$ | $\mathbf{58.7 \pm 3.6}$ | $62.5 \pm 1.9$ | $66.2 \pm 0.5$ | $\mathbf{16.2 \pm 1.3}$ | $17.9 \pm 1.0$ | $23.8 \pm 1.1$ | $30.2 \pm 0.4$ |
| pFedHN-sampled | $24.8 \pm 1.0$ | $\mathbf{61.0 \pm 3.7}$ | $60.5 \pm 0.9$ | $\mathbf{68.5 \pm 0.7}$ | $3.9 \pm 0.4$ | $13.5 \pm 0.5$ | $25.7 \pm 0.6$ | $\mathbf{32.4 \pm 0.4}$ |
| pFedHN-ensemble | $47.6 \pm 3.2$ | $\mathbf{62.2 \pm 3.7}$ | $63.6 \pm 1.0$ | $\mathbf{68.5 \pm 0.7}$ | $7.8 \pm 1.8$ | $20.4 \pm 1.2$ | $\mathbf{29.3 \pm 0.9}$ | $\mathbf{32.7 \pm 0.5}$ |
| pFedHN-nearest | $24.4 \pm 6.2$ | $\mathbf{63.1 \pm 3.48}$ | $58.4 \pm 2.6$ | $\mathbf{68.5 \pm 0.7}$ | $6.5 \pm 2.7$ | $14.3 \pm 0.6$ | $25.2 \pm 1.2$ | $\mathbf{32.1 \pm 0.6}$ |
| IT-PFL-HN (ours) | $\mathbf{55.9 \pm 2.3}$ | $61.3 \pm 4.5$ | $\mathbf{64.7 \pm 1.8}$ | $67.9 \pm 0.7$ | $9.9 \pm 2.0$ | $\mathbf{24.4 \pm 1.6}$ | $\mathbf{29.3 \pm 1.1}$ | $\mathbf{32.7 \pm 0.5}$ |

model; (2) **FedAVG** (McMahan et al., 2017a), perhaps the most widely used FL algorithm. All novel clients are evaluated using a single global model; (3) **pFedHN** (Shamsian et al., 2021), Currently the state-of-the-art approach to PFL. PFL-HN is based on a HN. Sine PFL methods produce a set of models, one for each training client, we tested three different ways to use training models for inference with the novel client. **(3a) FedHN-sampled:** Draw a trained client model uniformly at random. We evaluate this baseline by computing the mean accuracy of all personalized models on each novel client. **(3b) FedHN-Ensemble:** A stronger baseline is achieved by taking into account all personalized models when predicting for a single novel client. This is achieved by averaging the logits of all models for each prediction. In practice, this method requires sending multiple models to each novel client. Therefore, it is expensive in both communication and computation costs. **(3c) FedHN-nearest:** We measured the distance between a novel client and all trained clients, and use the model of the closest training client. Since in FL the data must not leave the client, common methods to measure divergence between datasets Instead, we use A-distance (Ben-David et al., 2007), which measures how hard it is to separate data points from two different clients using a linear model. Since that linear model can get gradients from each client separately, it can be used in the FL setup.

## 6.3 RESULTS ON CIFAR DATA

We evaluate IT-PFL-HN using CIFAR10 and CIFAR100 datasets (Krizhevsky et al., 2009). Since these datasets do not have a natural partitioning into clients, we follow two commonly used protocols to split the data across clients.

**(1) Pathological split:** As proposed by McMahan et al. (2017a), we sort the training samples by their labels and partition them into $N \cdot K$ shards. Each client is then randomly assigned $K$ of the shards. This results in $N$ clients with the same number of training samples and a different distribution over labels. In our experiments, we use $N = 100$ clients, $K = 2$ for CIFAR10 and $K = 5$ for CIFAR100.

**(2) Dirichlet allocation:** We follow the procedure by Hsu et al. (2019a) to control the magnitude of distribution shift between clients. For each client $i$, samples are drawn independently with class labels following a categorical distribution over classes with a parameter $q_i \sim Dir(\alpha)$. Here, $Dir$ is the symmetric Dirichlet distribution and $\alpha$ is the concentration parameter. We conduct three experiments for each of the two datasets with $\alpha \in 0.1, 0.5, 10$. Smaller values of alpha imply larger distribution shifts between clients.

**Implementation details:** There are three different models in IT-PFL-HN: A target model, a hypernetwork, and a client encoder. **Target model:** When evaluating using CIFAR10 and CIFAR100 we use a LeNet-based (LeCun et al., 1998) network with two convolutions and two fully connected layers. To assure a fair comparison, we use the same target model across all evaluated methods and baselines. **Client encoder:** Same architecture as the target model above but with an additional fully connected layer followed by Global-Max and Global-Mean operations over batch samples. These layers are added after each convolution layer and before the fully connected layers. The output dimension is the size of the embedding dimension rather than the number of labels. **Hypernetwork:** The hypernetwork is a fully connected neural network, with three hidden layers and multiple linear heads per target weight tensor as in Shamsian et al. (2021).

**Results:** Table 1 shows IT-PFL-HN performs better than or equivalent to the evaluated FL and PFL methods in all evaluated datasets and split, except for CIFAR-100 pathological split.

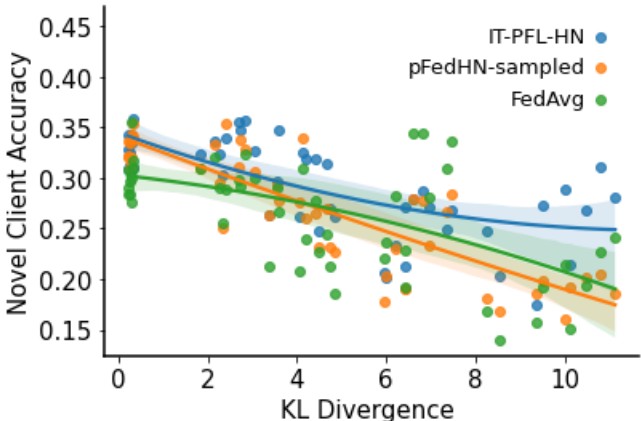

Figure 3: **Accuracy of novel clients vs. distribution shift between the novel client and training clients**. Shown results for CIFAR-100 across multiple splits to $N_{train} = 90$ train clients and $N_{novel} = 10$ novel clients using symmetric Dirichlet distributions with varying parameter $\alpha$ (see 6.5). For each test client, we calculate its KL-divergence to the nearest train client. Accuracies of novel clients are reported against the KL-divergence from the nearest train client for each method.

## 6.4 REAL-WORLD DATA

We next extend our experiments to more realistic datasets with data splits across clients that simulate real-world distributions. We use two different datasets studied in Hsu et al. (2020). **(1) iNaturalist:** A dataset for Natural Species Classification based on the iNaturalist 2017 Challenge (Horn et al., 2018). The dataset has 1,203 classes. **(2) Landmarks:** A dataset for landmark recognition based on the 2019 Landmark Recognition Challenge (Weyand et al., 2020). The dataset has 2,028 classes. We split the datasets into clients following the splits suggested by Hsu et al. (2020). We evaluate IT-PFL-HN using two of the suggested geographical splits of iNaturalist: iNaturalist-Geo-1k and iNaturalist-Geo-300 with 368 and 1,208 users respectively, and using the proposed split by authorship of Landmarks, Landmarks-User-160k, with 1,262 users.

**Implementation details:** We first use a MobileNetV2 model (Sandler et al., 2018) pre-trained on ImageNet (Deng et al., 2009) to extract features for each image. The extracted feature vectors, of length 1280, are the input for both the target model and the client encoder. **Target model:** When evaluating iNaturalist and Landmarks datasets the target model is a simple fully connected network with two Dense layers and a Dropout layer. We use the same target model across all evaluated methods and baselines. **Client encoder:** the client encoder architecture has three fully connected layers with Global-Max and Global-Mean operations over batch samples after the first layer. The output dimension is the size of the embedding dimension. **Hypernetwork:** The HN is a fully connected neural network, with three hidden layers and multiple linear heads per target weight tensor as in the previous experiments.

**Results:** Table 2 shows IT-PFL-HN outperforms current FL and PFL methods in both evaluated splits of the iNaturalist dataset. IT-PFL-HN ties with the pFedHN - ensemble baseline on the Landmarks dataset. However, pFedHN - ensemble suffers from relatively large communication and computation costs compared to the proposed IT-PFL-HN.

Table 2: **Accuracy on novel Clients, Real-World Data:** Values are averages and standard error across clients.

|  | iNaturalist | | Landmarks |
|---|---|---|---|
| split | Geo-300 | Geo-1k | User-160k |
| FedAvg | $36.14 \pm 1.60$ | $36.89 \pm 1.12$ | $34.85 \pm 1.28$ |
| pFedHN - sampled | $25.58 \pm 1.39$ | $27.23 \pm 0.86$ | $37.39 \pm 1.31$ |
| pFedHN - ensemble | $31.54 \pm 1.65$ | $36.55 \pm 1.16$ | $\mathbf{39.08 \pm 1.38}$ |
| pFedHN - nearest | $24.16 \pm 3.65$ | $26.92 \pm 4.55$ | $33.06 \pm 3.56$ |
| IT-PFL-HN (ours) | $\mathbf{37.47 \pm 1.65}$ | $\mathbf{41.57 \pm 1.17}$ | $\mathbf{39.4 \pm 1.43}$ |

## 6.5 EFFECT OF DISTRIBUTION SHIFT

To measure the effect of distribution shift on the results we synthesize additional splits of the CIFAR-100 dataset using the same Dirichlet allocation procedure described in section 6.3. We evaluate three methods: FedAvg, pFedHN-sampled and IT-PFL-HN, while varying $\alpha \in \{0.1, 0.25, 0.5, 1, 10\}$. To

Table 3: Test mean accuracy (±SEM) for corrupted CIFAR-10 novel clients.

|  | Blur | Rotation | Brightness | Contrast | Saturation |
|---|---|---|---|---|---|
| FedAvg | 42.57±0.02 | 22.28±0.03 | 21.94±0.03 | 21.10±0.03 | 47.48±0.03 |
| pFedHN - sampled | 22.99 ±0.01 | 13.23±0.02 | 13.37±0.02 | 12.80±0.02 | 24.73±0.01 |
| pFedHN - ensemble | 38.00 ±0.04 | 21.76±0.06 | 21.75±0.06 | 21.43±0.06 | 43.70±0.04 |
| pFedHN - nearest | 8.37 ±0.05 | 11.70±0.10 | 11.99±0.09 | 11.16±0.04 | 20.70±0.11 |
| IT-PFL-HN (ours) | **46.17±0.01** | **23.34±0.05** | **22.57 ±0.04** | **22.08±0.04** | **53.45±0.02** |

estimate the "distance" between a novel client and the training clients we measure the KL-divergence between the novel client and the nearest trained client. The KL-divergence is computed over the empirical distributions of labels.

Figure 3 presents the accuracy of the novel client as a function of its Kl-divergence to the nearest train client. As Expected, the accuracy over novel clients decreases as the KL-divergence between that novel client and the training clients grows. This holds for all three evaluated methods. IT-PFL-HN method demonstrates the most moderate decrease in performance as KL-divergence grows, and achieves the highest accuracy for novel clients with large distribution shifts.

### 6.6 IT-PFL-HN ON NOISY DATA

In real life, many times the domain shift of the novel client is caused by using different sensors and/or different environments. We evaluated IT-PFL-HN robustness to five common sensor and environment corruptions: blur, rotation, brightness, contrast, and saturation.

We evaluate this domain shift using CIFAR-10. The data from training clients were kept noncorrupted. The data of the novel client was corrupted in five different ways: *Blur*, *Rotation*, *Brightness*, *Contrast*, *Saturation* For blur, we used a $7 \times 7$ Gaussian filter, with sigma that is chosen uniformly at random in the range of (0.1, 2.0). For brightness, contrast, and saturation, we use a factor that is chosen uniformly at random in the range of (0.5, 1.5). The rotation transformation used an angle that was chosen uniformly at random to lie in the range of (0, 15).

Table 3 summarizes the results. IT-PFL-HN outperforms the evaluated FL and PFL baselines when evaluated on corrupted novel clients.

## 7 DIFFERENTIAL PRIVACY

A key aspect of federated learning is data privacy, restricting clients from sharing their data directly with the hub. Unfortunately, since data is used to train the federated model, and the federated model produces models for other clients, some private information may be exposed (See a recent survey by Mothukuri et al. (2021)). In this section, we analyze the privacy of a novel client and characterize how it can protect its privacy by applying differential privacy (DP). We further show the trade-off between the privacy of a novel client and the accuracy of the personalized model.

We first define key concepts and our notation. Following Abadi et al. (2016), we say that two datasets $D, D'$ are adjacent if they differ in a single instance. A randomization mechanism $M : D \to R$ satisfies a $(\epsilon, \delta)$-differential privacy if for any two adjacent inputs $d, d' \in D$ and for any subset of outputs $S \subseteq R$ it holds that $Pr[M(d) \in S] \leq e^\epsilon Pr[M(d') \in S] + \delta$. Here, $\epsilon$ quantifies privacy loss - smaller values mean better privacy protection and $\delta$ bounds the probability of privacy breach. Finally, given two datasets $D$ and $D'$ that differ in only one element, the sensitivity of a function $f$ is defined by $\Delta f = \max_{D,D'} ||f(D) - f(D')||$.

Dwork et al. (2006) showed that given a model $f$, data privacy can be preserved by perturbing the output of the model, and calibrating the standard deviation of the noise according to the sensitivity of the function $f$, and the desired level of privacy $\epsilon$. Intuitively, if the protected model is not very sensitive to changes in a single training element, one can achieve DP with smaller perturbations.

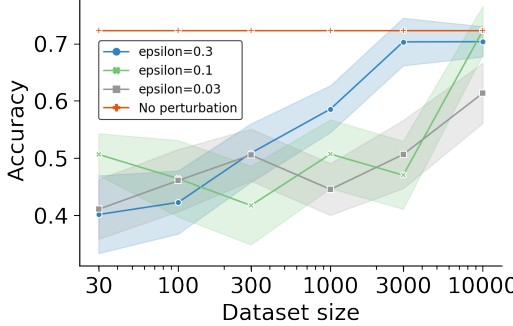

Figure 4: Test accuracy ($\pm$SEM) for CIFAR-10 novel client while applying DP. As the dataset size increases, we add less noise to obtain the same privacy level.

Our focus here is to apply DP to IT-PFL novel client. Fortunately, the only information that a novel client shares is the embedding. It is computed locally by the client, so applying a DP procedure to the encoder can protect the data privacy of the novel client. As a side benefit, the novel client applies the encoder at the inference time, not affecting the training convergence time and accuracy.

Several mechanisms were proposed for noise adding to achieve DP. Here we focus on the mechanism described by Abadi et al. (2016), but the same method can be used with other mechanisms. Abadi et al. (2016) describes a Gaussian mechanism that adds noise drawn from Gaussian distribution with $\sigma^2 = \frac{2\Delta f^2 log(1.25/\delta)}{\epsilon^2}$.

Let a novel client apply $(\epsilon, \delta)$-DP to the encoder using the Gaussian mechanism. The server sends to the encoder $g$ to the client. Then the client send to the server $g(\{x_j^i\}_{j=1}^{m_i}) + \xi$ as its embedding, where $\xi$ is an IID vector from Gaussian distribution with $\sigma^2 = \frac{2\Delta g^2 log(1.25/\delta)}{\epsilon^2}$. To do that, the client has to know what is the encoder sensitivity. The following lemma shows that when using a deep-set encoder, we can bound the sensitivity of the encoder, hence bound the noise magnitude necessary to achieve privacy. See proof in appendix D.

**Lemma 7.1.** *Let $g$ be a deep-set encoder, written as: $g(D) = \psi(\frac{1}{|D|}\sum_{x \in D} \phi(x))$. If $\psi$ is a linear function with Lipschitz constant $L_\psi$, and $\phi$ is bounded by $B_\psi$, then the sensitivity of the encoder is bounded by $\Delta g \leq \frac{2}{|D|} L_\psi B_\phi$.*

The lemma shows that the sensitivity of the encoder decreases linearly with the size $|D|$ of the novel client dataset. For a given $(\epsilon, \delta)$, lower sensitivity means lower Gaussian noise, which leads to better performance.

We now evaluate empirically the effect of adding noise to the embedding of a novel client. To meet the conditions in lemma 7.1, we normalized the output of $\phi$ to be on a unit sphere, so $B_\phi = 1$. In addition, we use a simple average on the output of $\phi$, so $L_\psi = 1$. We used $\delta = 0.01$ and compare different values of $\epsilon$ and dataset size.

Figure 4 shows that with sufficient data ($n \geq 3000$ for $\epsilon = 0.3$), a novel client can protect its privacy without compromising the performance of the personalized model.

## 8 CONCLUSION

This paper describes IT-PFL, a new real-world federated setup, focusing on transferring a model trained in a feerated learning manner, to novel clients that were not available during training, and dont even have labeled data.

We propose IT-PFL-HN, a novel approach for IT-PFL, based on learning an encoder that learns a space of clients, and a hypernetwork that can map a client to its corresponding model in an "on-demand" way. We evaluated IT-PFL-HN on four benchmark datasets, showing that it usually generalizes better than current FL and PFL methods. Finally, we analyze and bound the generalization error for the novel client and show that our approach can guarantee differential privacy. We hope this paper will encourage the research community to consider generalization to novel clients when designing FL methods.

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

## A  GENERALIZATION BOUND

In the context of MTL and DA, the "vanilla" supervised learning setup, can be viewed as if we have only one task and one domain. The most common solution is to find $h \in \mathcal{H}$ that minimizes the loss function on a training set sampled from probability distribution $P$. In general, $\mathcal{H}$ is a hyperparameter, defined by the network architecture. This Blumer et al. (1989) shows that in this simple case, the generalization error is bounded. The bound depends upon the "richness" of $H$. Choosing a "rich" $H$ (with large VC-dimension), let the generalization error be larger.

In IT-PFL, each client may use a different hypothesis. Instead of bounding the error of one chosen hypothesis, we bound the error of the chosen hypothesis space that each client chooses from. This way, we can bound the error of a novel client, without assuming anything about the way it chose from the hypothesis space.

First, we find $H$ that contains hypotheses that can fit all data of the clients. Second, for each client, we select the best hypothesis $h \in H$ according to the client data. We define $Q$ as a distribution over $P$, so, each client sample from $Q$ the distribution $P_i$. We further define $\mathbb{H}$ as a hypothesis space family, where each $H \in \mathbb{H}$ is a set of functions $h : X \to Y$.

The first goal is to find a hypothesis space $H \in \mathbb{H}$ that minimizes the weighted error of all clients, assuming each client uses the best hypothesis $h \in H$. We define this error using the following loss:

$$err_Q(H) := \int_P \inf_{h \in H} err_P(h) dQ(P) \tag{3}$$

while $err_P(h) := \int_{X \times Y} l(h(x), y) dP(x, y)$. In practice $Q$ in unknown, so we can only estimate $err_Q(H)$ using the sampled clients and their data.

For each client $i = 1..n$, we sample the client training data from $X \times Y \sim P_i$. We denote the sampled training set with $z_i := (x_1, y_1), ..., (x_m, y_m)$, and $z = z_1, ...z_n$. The empirical error of a specific hypothesis is defined by $\hat{er}_z(h) := \frac{1}{m} \sum_{i=1}^{m} l(h(x_i), y_i)$. Now the empirical loss to minimize is

$$\hat{er}_z(H) := \frac{1}{n} \sum_{i=1}^{n} \inf_{h \in H} \hat{er}_{z_i}(h) \tag{4}$$

Baxter (2000) shows that if the number of clients n satisfies $n \geq max\{\frac{256}{\epsilon^2} log(\frac{8C(\frac{32}{\epsilon}, H^*)}{\delta}), \frac{64}{\epsilon^2}\}$, and the number of samples per client $m$ satisfies $m \geq max\{\frac{256}{n\epsilon^2} log(\frac{8C(\frac{32}{\epsilon}, H_l^n)}{\delta}), \frac{64}{\epsilon^2}\}$, then with probability $1 - \delta$ all $H \in \mathcal{H}$ satisfies

$$err_Q(H) \leq \hat{er}_z(H) + \epsilon \tag{5}$$

were $C(\frac{32}{\epsilon}, H_l^n)$ and $C(\frac{32}{\epsilon}, H_l^n)$ are the covering numbers defined in Baxter (2000), and can be referred as a way to measure the complexity of $H$. Note that a very "rich" $H$ makes $\hat{er}_z(H)$ small, but it increases the covering number, so for the same amount of data, $\epsilon$ increases.

For the PFL setup, this is enough, since we can ensure that for a client that sampled from $Q$ and was a part in the federation, the chosen hypothesis $h \in H$ has an error close to the empirical one $\hat{er}_z(H)$. For a novel client, this may not be the case. The novel client may sample from a different distribution over P. In the general case, the novel client may even have a different distribution over $X \times Y$. In the most general case, the error on the novel client can not be bound. In DA, a common distribution shift is a covariate shift, where $P(x)$ may change, but $P(x|y)$ remains constant. This assumption lets us bound the error of the novel client.

Ben-David et al. (2010) proofed that for a given $H \in \mathcal{H}$, if S and T are two datasets with $m$ samples, then with probability $1 - \delta$, for every hypothesis $h \in H$:

$$err_T(h) \leq err_S(h) + \frac{1}{2}\hat{d}_{H\Delta H}(S, T) + 4\sqrt{\frac{2d \, log(2m) + log(\frac{2}{\delta})}{m}} + \lambda \tag{6}$$

where $err_D(h) = E_{(x,y) \, D}[|h(x) - y|]$ is the error of the hypothesis on probability distribution of the domain $D$. $\hat{d}_{H\Delta H}(S, T)$ is a distance measure between the domains S and T, and $\lambda =$

$\arg\max_{h \in H} err_T(h) + err_S(h)$. Note that for over-parametrized models like deep neural networks $\lambda$ should be very small. To keep the analysis shorter we assume this is the case. We further assumed $m$ is big enough to neglect $4\sqrt{\frac{2d \, log(2m) + log(\frac{2}{\delta})}{m}}$. Those assumptions are not mandatory, and the following analysis can be done without it.

## B    MORE DETAILS ABOUT TRAINING

In IT-PFL-HN two main components are trained using a federation of labeled clients: A hypernetwork and a client encoder. The hypernetwork optimizes the $L_{HN}$ loss defined in 7 by updating both its own weights $\theta$ and the clients representations $\{e_k\}_{k=1}^N$.

$$L_{HN}(\theta, e_1, ..., e_N) = \sum_{i=1}^{n} \sum_{j=1}^{m_i} l(f_\theta(e_i)(x_j^i), y_j^i) \tag{7}$$

The client encoder trains to predict the representations learned by the hypernetwork from the client raw data by minimizing $L_{encoder}$ defined in 8. At inference time, a novel client feeds its data to the client encoder and gets an embedding vector. Then, feeding the embedding vector to the hypernetwork produces a custom model for the client.

$$L_{encoder} = \sum_{i=1}^{n} L_2(g_\gamma(\{x_j^i\}_{j=1}^{m_i}), e_i) \tag{8}$$

In detail, in each communication step, the server selects a random client. Using its current embedding, the hypernetwork generates a customized network and communicates it to the client. The client then locally trains that network on its data for a predefined number of local epochs. As in Shamsian et al. (2021), the client communicates the delta between the weights before and after the training back to the server. Using the chain rule, the server can train the hypernetwork and the embeddings to optimize $L_{HN}$ (see Figure 2).

In addition to the custom target model, the server sends the client the current encoder, and the current embedding of the client. Similar to the previous step, the client trains the encoder locally to predict the embedding from the client data by optimizing $L_{encoder}$, then, the updates of the encoder are sent back to the server for aggregation.

Up to this point, the client encoder trains in parallel to the hypernetwork, and has no influence on the hypernetwork weights or the embeddings of the labeled clients. We found that freezing the encoder and fine-tuning the hypernetwork using the trained encoder predictions improve the results of our method. This is done by optimizing the hypernetwork parameters $\theta$ using $L_{Fine-tune}$.

$$L_{Fine-tune}(\theta) = \sum_{i=1}^{n} \sum_{j=1}^{m_i} l(f_\theta(g_\gamma(\{x_j^i\}_{j=1}^{m_i}))(x_j^i), y_j^i) \tag{9}$$

However, this fine-tune step reduces the performance of the labeled clients. Since our goal is to generalize well to novel clients, we ignored this effect. Note that in a real-world application, the server may save a version of the hypernetwork before fine-tuning it, and used it when generating models for the original federation.

## C    EXPERIMENTAL DETAILS

For all experiments presented in the main text, we use a fully-connected hypernetwork with 3 hidden layers of 100 hidden units each. The size of the embedding dim is $\frac{N_{clients}}{4}$. Experiments are limited to 500 communication steps. In each step, communication is done with $0.1 \cdot N_{train}$.

**Hyperparmeter Tuning**    We divide the training samples of each train client into 85% / 15% train / validation sets. The validation sets are used for hyperparameter tuning and early stopping of all baselines and datasets. The searched hyperparameters and corresponding values by method:

**FedAVG**: The local momentum $\mu_{local}$ is set to 0.5. We search over local learning-rate $\eta_{local} \in \{1e-1, 5e-2, 1e-2, 5e-3, 1e-3\}$, number of local epochs $K \in \{1,,2,5,10\}$ and batch size $\{16, 32, 64\}$. **pFEdHN**: We set $\mu_{local} = 0.9$. We search over learning-rates of the hypernetwork, embedding layer and local training: $\eta_{hn}, \eta_{embedding}, \eta_{local} \in \{1e-1, 5e-2, 1e-2, 5e-3, 1e-3\}$, weight decays $wd_{hn}, wd_{embedding}, wd_{local} \in \{1e-3, 1e-4, 1e-5\}$, number of local epochs $K \in \{1,,2,5,10\}$ and batch size $\{32, 64\}$. **IT-PFL-HN**: We perform the optimization using the same parameters and values as in pFEdHN. In addition, we search over the learning-rate of the client encoder $\eta_{encoder} \in \{1e-1, 5e-2, 1e-2, 5e-3, 1e-3\}$.

**CIFAR(Section 6.3)** We use a LeNet-based target network with two convolution layers with 16 and 32 filters of size 5 respectively. Following these layers are two fully connected layers of sizes 120 and 84 that output logits vector. The client encoder follows the same architecture with an additional fully connected layer of size 200 followed by Mean-global-pooling for the first 100 units and Max-global-pooling for the other 100 units. Global-pooling is done over the samples of a batch.

**Real-World Data(Section 6.4)** We use a simple fully-connected network with two Dense layers of size 500 each, followed by a Dropout layer with a dropout probability of 0.2. The client encoder is a fully-connected network with three Dense layers of size 500. The first layer is followed by Mean-global-pooling for the first 250 units and Max-global-pooling for the other 250 units.

## D   DIFFERENTIAL PRIVACY

*Proof.*

$$\Delta g := \max_{D,D'} ||g(D) - g(D')|| = \max_{D,D'} ||\psi(\frac{1}{|D|}\sum_{x \in D}\phi(x)) - \psi(\frac{1}{|D'|}\sum_{x \in D'}\phi(x))||. \quad (10)$$

Denote $d \in D$ and $d' \in D'$ as the only nonidentical instance between $D$ and $D'$, so $D/d = D'/d'$. Then

$$\Delta g = \max_{D,D'} ||\psi\left(\frac{1}{|D|}[\sum_{x \in D/d}\phi(x) + \phi(d)]\right) - \psi\left(\frac{1}{|D'|}[\sum_{x \in D'/d'}\phi(x) + \phi(d')]\right)|| \quad (11)$$

$$= max_{d,d'}\frac{1}{|D|}||\psi\left(\phi(d) - \phi(d')\right)|| \quad (12)$$

Assume that $\phi$ is bounded with $B_\phi$, so $|\phi(d) - \phi(d')| < 2B_\phi$. Then from the linearity of $\psi$:

$$\Delta g \leq \frac{1}{|D|}L_\psi|\phi(d) - \phi(d')| \leq \frac{2}{|D|}L_\psi B_\phi \quad (13)$$

$\square$

