# OpenReview forum: "Inference-Time Personalized Federated Learning"
_ICLR.cc/2022/Conference — ICLR 2022 Submitted_

### Official Review · Reviewer_GyWf · 2021-10-30

**Correctness:** 3
**Technical Novelty And Significance:** 2
**Empirical Novelty And Significance:** 3
**Recommendation:** 5
**Confidence:** 4

**Main Review:**

This paper is clearly written. The authors adopt an additional novel client to enhance the accuracy of personalized FL. However, the reviewer still has several concerns:
1. The proposed generalization bound is established for the novel client. How does this generalization-bound reflect the efficacy of the proposed personalized federated learning framework?
2. The compared baselines are not sufficient to demonstrate the efficacy of the proposed algorithm. In fact, there exist several PFL algorithms, such as Ditto, Sub-FedAvg, etc.
3.  The authors should do an additional ablation study to evaluate the influence of the size of the unlabeled dataset.  It seems that is the novel client merely has limited data. The proposed algorithm in this work reduces a smile variant of pFedHN.


**Summary Of The Paper:**

The authors propose a new personalized federated learning paradigm composed of a hypernetwork module and an encoder module in server and an extra novel client with unlabeled data.  The encoder module is enhanced thanks to the unlabeled data.   Preliminary experiments demonstrate the efficacy of the proposed algorithm.

**Summary Of The Review:**

see the comment above.

---

### Official Review · Reviewer_pXWe · 2021-10-30

**Correctness:** 4
**Technical Novelty And Significance:** 1
**Empirical Novelty And Significance:** 2
**Recommendation:** 3
**Confidence:** 4

**Main Review:**

While the authors claimed that IT-PFL is a novel problem, the solution that the authors gave is similar to [1], which utilizes hypernetwork in personalized federated learning task. At least to me, the difference between [1] and this paper is marginal, which makes me question the novelty of the paper. I think the authors need to make clear what is the novel contribution given the existence of [1]. This paper does give some results on generalization bound and differential privacy, but I believe they are just results of simple application of existing theorems. The theoretical contributions from these two sections are not strong enough, or at least the authors did not make clear what are theoretical challenges.

Besides the novelty concern, I have two additional suggestions.

First, I think it will be interesting to compare hypernetworks and meta-learning. These two approaches are solving essentially the same problem in the end, and they are both applied to personalized federated learning. A deep understanding of the connections and differences between the two approaches will be beneficial to the community.

Second, I think in related work, the authors should summarize the literature about Hypernetworks (HNs), and especially its recent application in personalized federated learning ([1]), because there is obviously an intimate connection between that line of works and this paper.

[1] Aviv Shamsian, Aviv Navon, Ethan Fetaya, and Gal Chechik. Personalized federated learning using hypernetworks. arXiv preprint arXiv:2103.04628, 2021.

**Summary Of The Paper:**

The paper proposes a new task named as Inference-Time Personalized Federated Learning (IT-PFL). Specifically, given a new client with unlabeled data joining in the federated learning system after the training process has finished, IT-PFL aims to deploy personalized FL model to it. The strategy is based on recent works on hypernetwork. The training strategy is in an end-to-end framework.

**Summary Of The Review:**

Given the major novelty concern, I do not think the paper is appropriate for acceptance in its current shape. The authors should try to make clear what is the novel part of the paper, how challenging it is to get those results, and add more literature review in related work.

---

### Official Review · Reviewer_MpJV · 2021-11-01

**Correctness:** 3
**Technical Novelty And Significance:** 2
**Empirical Novelty And Significance:** 3
**Recommendation:** 5
**Confidence:** 3

**Details Of Ethics Concerns:**

The federated learning is highly relevant to the privacy aspects. This work presents a new problem where training is on other clients and inference on new client. It would be good to do ethical reviews.


**Main Review:**

Pros:
1. The paper overall in clearly written and easy to follow. The overall flow of the paper and descriptions of the proposed approach is clear.
2. The problem being study is well-motivated and can be useful.
3. It also provide differential privacy analysis to provide insights into the privacy protection perspective of the proposed approach.

Cons:
1.  Some of the experiment results are not explained. For example, in Table one, the CIFAR-100 pathological split, the proposed approach is not outperforming the baseline approaches. However, it lacks any explanation or analysis about why it is the case.
2. Question: in the real-world dataset, for the models both target model, client encoder, and hyper network. These are all simple fully connected learns with global operations. Are these models realistic for real-world use-cases? If these base model change, will the framework performance differ? It would be good to demonstrate the effectiveness of the proposed approach with different types of neural networks.
3. In the experiment, Section 6.4, it methods the pFedHN-ensemble suffers from large communication and computation cost. However, there are not numbers supporting the claim. It would be good to provide numbers to see if these are real-concerns.
4. I was wondering how much difference it will make in comparison with the new client participate in training. It would be good to add experiments to compare with that.

Minors:
1. In section 4, Meta mechanism, the quote is not right. same for many other quotes

**Summary Of The Paper:**

This paper studies the problem of in personalized federated learning, the current paradigm does not allow for new clients to join during the inference time. Federated learning does not generalize well to new data distribution that is very different from training. Personalized FL are not designed to apply to a client that is not being trained. This paper defines a new task inference-time personalized federated learning (ITPFL) to close the gap. The proposed approach IT-PFL-HN first trains a client encoder, then maps a full client dataset to descriptor. A hyper network maps the descriptors on to a space. During inference time, the client description can be computed locally and request the personalized model on-demand. The paper demonstrate the effectiveness of the proposed approach on both CIFAR and real-world datasets.


**Summary Of The Review:**

Overall, the problem being studied in this paper is interesting and well-motivated. The paper is clearly written. The novelty perspective of this paper can be highlighted better. It seems like a lot of the aspect like hyper network are proposed by other approaches.  Some other major concerns are in terms of the experiments, there are results are not explained or claims without number supported. Please see main review for details.

---

### Official Review · Reviewer_o4oc · 2021-11-02

**Correctness:** 3
**Technical Novelty And Significance:** 2
**Empirical Novelty And Significance:** 2
**Recommendation:** 5
**Confidence:** 4

**Main Review:**

Strength: The proposed idea contributes to the hypernetwork research and ourperforms current state-of-the-art results.

Weaknesses: it's clear how the proposed paper is different from [1] but I would dedicate more space for the comparison, specifically Section 5.3 of [1] suggests using the nearest client (as you describe in Table 1). It's clear that CIFAR-100 or iNaturalist have more diverse data where your method excels, but I wonder if in real scenario with millions of users participating (e.g. language model for Reddit/Stackoverflow dataset) the same conclusion would persist.

It would be useful to see the distribution and not the average accuracy values for new user performance. It might be that the proposed method benefits (or harms) the users with diverse data, which is an essential question for FL fairness.

It leads to the following point: evaluating the method on the text data is important for FL as language modeling is one of the few industry-deployed use cases and widely used in FL literature. And also the personalization of LM models is an important problem.

For DP please clarify what d \in D is? Is it a single input of the user?

Lastly, other personalization works that use local adaptation [2], such as finetuning global model on the local data, should be considered as baselines and I wonder what is their performance w.r.t. the proposed method.

[1] Shamsian, A., Navon, A., Fetaya, E., & Chechik, G. "Personalized Federated Learning using Hypernetworks." In ICML'21
[2] Li, Tian, Shengyuan Hu, Ahmad Beirami, and Virginia Smith. "Ditto: Fair and robust federated learning through personalization." In ICML'21.

**Summary Of The Paper:**

This paper proposes to use hypernetwork to personalize the federated model by encoding new user data and using this embedding as a parametrization argument. The results demonstrate significant average improvement over the new clients. Furthermore, the authors evaluate the possible use of DP to encode the user embedding vectors.

**Summary Of The Review:**

The paper is well-written and presents a new algorithm that uses encoder to adapt for new clients. The paper lacks few baselines and explanations but can be beneficial for FL community.

---

### Decision · Program_Chairs · 2022-01-20

**Decision:**

Reject

**Comment:**

This paper proposes a personalized federated learning method using a hyper-network to encode unlabeled data from new clients. At inference time, new clients can use unlabeled data as input to this hyper-network in order to obtain a personalized version of the model. The key strength of the paper is that the idea is interesting and timely. Personalization has been studied for clients that participate from the beginning of training, but personalization of models for new clients that join later on has not been considered in most previous works. The experimental results also show a reasonable improvement over the baselines. However, the following concerns remain:
1) Novelty in comparison with reference [1]. Please add a detailed comparison when you revise the paper.
2) Explanation of the experimental results and comparison with baselines was deemed insufficient by some of the reviewers.
3) The generalization bound and the DP results seem standard extensions of existing works and do not add much novelty to the paper.

There wasn't much post-rebuttal discussion and the reviewers decided to stick to their original scores. Therefore, I recommend rejection of the paper. I hope that the authors will take the reviewers' constructive comments into account when revising the paper for a future resubmission.